# Dietary Components, Microbial Metabolites and Human Health: Reading between the Lines

**DOI:** 10.3390/foods9081045

**Published:** 2020-08-03

**Authors:** Yao Guo, Xiaohan Bian, Jiali Liu, Ming Zhu, Lin Li, Tingyu Yao, Congjia Tang, Vinothkannan Ravichandran, Peng Liao, Konstantinos Papadimitriou, Jia Yin

**Affiliations:** 1Hunan Provincial Key Laboratory of Animal Intestinal Function and Regulation, College of Life Sciences, Hunan Normal University, Changsha 410006, China; guoyao0807@hotmail.com (Y.G.); bianxhbio@163.com (X.B.); 18711175173@163.com (J.L.); ZhuMing96@smail.hunnu.edu.cn (M.Z.); lilin191011@163.com (L.L.); yaotingyu102895@163.com (T.Y.); tcj1670403118@163.com (C.T.); 2Hunan International Joint Laboratory of Animal Intestinal Ecology and Health, College of Life Science, Hunan Normal University, Changsha 410006, China; 3State Key Laboratory of Microbial Technology, Shandong University–Helmholtz Institute of Biotechnology, Shandong University, Qingdao 266237, China; vrvinothan@sdu.edu.cn; 4Institute of Subtropical Agriculture, Chinese Academy of Sciences, Changsha 410125, China; liaopeng@isa.ac.cn; 5Department of Food Science and Technology, School of Agriculture and Food, University of Peloponnese, 22131 Antikalamos, Greece; kostas.papadimitriou@gmail.com

**Keywords:** carbohydrates, amino acids, lipids, eating habits, human health

## Abstract

Trillions of bacteria reside in the human gut and they metabolize dietary substances to obtain nutrients and energy while producing metabolites. Therefore, different dietary components could affect human health in various ways through microbial metabolism. Many such metabolites have been shown to affect human physiological activities, including short-chain fatty acids metabolized from carbohydrates; indole, kynurenic acid and para-cresol, metabolized from amino acids; conjugated linoleic acid and linoleic acid, metabolized from lipids. Here, we review the features of these metabolites and summarize the possible molecular mechanisms of their metabolisms by gut microbiota. We discuss the potential roles of these metabolites in health and diseases, and the interactions between host metabolism and the gut microbiota. We also show some of the major dietary patterns around the world and hope this review can provide insights into our eating habits and improve consumers’ health conditions.

## 1. Introduction

In recent years, many researchers studied the gut-microbiota to understand how it influences our health. Through mass spectrometry and 16S rRNA sequencing, researchers found that microbial metabolism could affect the chemical profile of various organs, especially the gut [1]. Dietary components such as carbohydrates, proteins, and lipids are ingested by gut microbiota and the metabolites derived therein have been reported to influence the human health either directly or indirectly.

As for carbohydrates, most of them are water-soluble, like fructose, sucrose, and inulin (Figure 1). But polysaccharides are insoluble, like resistant starch, which cannot be hydrolyzed by endogenous enzymes in the small intestine, but it can be fermented in the colon and then be utilized by gut microbiota [2]. Cellulose is an insoluble fiber and is an abundant element of vegetarian diet. In spite of the fact that the fermentation of cellulose is limited, it could substantially change the colonic microbial composition [3].

The catabolism of aromatic amino acids (AAA) has been studied elaborately. Indole, serotonin, 3-indolepropionic acid (IPA) and para-cresol (*p*-cresol) are derived from AAA (Figure 1). Indole and serotonin act as signaling molecules, and drugs targeting the latter have been widely used in neuroscience [4,5]. *p*-cresol level increases in children with autism spectrum disorder, suggesting that it could be used as a biomarker [6].

Lots of daily foods are rich in lipids, especially meat, dairy products, and animal offal. Gut microbiota can metabolize lipids, such as cholesterol, phospholipids, plasma lipoproteins and glycerol, etc. into conjugated fatty acids (CLA), lipopolysaccharides, sheath phospholipids, ethanol amine phosphate and so on (Figure 1). Moreover, there are important connections between gut microbiota and some diseases associated with lipids, such as cardiovascular diseases and obesity [7].

Here, we show how some of the food elements are metabolized by gut microbiota and highlight some metabolites that have not been fully studied yet and we also cover some dietary patterns across the world. We hope this review will help people to realize the functions and importance of natural metabolites from gut microbiota, thus helping to improve people’s eating habits and public health.

## 2. Dietary Carbohydrates and Their Metabolites

### 2.1. Fructose

Fructose is one of the three dietary monosaccharides that is absorbed into the bloodstream straight away during digestion, along with glucose and galactose. Specifically, if the fructose is ingested as sucrose, sucrose will be degraded to free glucose and fructose by the catalysis of sucrose when it reaches the small intestine, and then each unit of glucose and fructose can be assimilated directly by the intestine (Figure 1).

A recent study shows that most of the fructose in the diet is metabolized in the small intestine [9]. Only when consuming low levels of fructose, the small intestine is able to clear and transform them into glucoses and organic acids. The gut is incapable of metabolizing high doses of fructose, so the excess fructose overflows into the liver and colon microbiota [9].

Although fructose enhances glucose metabolism [10], excessive intake of fructose may lead to increased insulin resistance, elevated low density lipoprotein, cholesterol and triglycerides, thus contributing to metabolic syndrome [11] and cardiovascular disease [12].

### 2.2. Sucrose, Xylooligosaccharides, and Fructooligosaccharides

Sucrose is a disaccharide molecule composed of glucose and fructose. It has been shown that feeding rats with sucrose induced obesity and increased the proportion of intestinal *Enterobacteriales*, especially *Escherichia coli* [13]. Sucrose is also associated with obesity and diabetes experimentally, clinically and epidemiologically [14].

The other two important polysaccharides are xylooligosaccharides (XOS) and fructooligosaccharides (FOS). XOS, polymers of the xylose, are produced from xylan fraction in plant fiber, and they have been used as prebiotics [15]. XOS are found in bamboo roots, fruits, vegetables, milk, and honey. Certain *Bacteroides* gut strains have evolved a complex strategy for the complete degradation of xylose oligosaccharides [16]. It has been experimentally proved that people drinking XOS-rich porridge for 6 weeks significantly increased the number of *Lactobacillus* spp. and *Bifidobacterium* spp. and decreased *Clostridium perfringens* without changing the total number of anaerobic bacteria [17]. Polysaccharide Utilization Loci in *Bacteroides* strains are gene clusters encoding diverse proteins, which have a variety of functions such as carbohydrate binding, transport and hydrolysis [16]. XOS could be degraded after the expression of these genes, thus producing SCFAs such as acetate, propionate, and butyrate (Figure 1) (Table 1).

SCFAs play a very important role in maintaining intestinal health. Firstly, they maintain intestinal acid–base balance. Studies have shown that the deficiency of SCFAs in the intestinal tract is related to increased pH in the intestinal environment. Such a high pH environment will stimulate the activities of bacterial tryptophanase and tyrosine dehydrogenase, thus leading to the increase in indole sulfate and *p*-cresol in the intestinal tract, which are harmful to intestinal health. Furthermore, it is reported that SCFAs participate in maintaining the integrity of the intestinal barrier, thus reducing the transfer of intestinal metabolic toxins to the circulatory system [26,27]. In addition, a recent study shows that SCFAs bind to the receptor GPR43/GPR109A, which transmits “hypertension signal”, thus directly affecting blood pressure and cardiovascular health [28].

FOS can be produced from fructan hydrolysis or from the catabolism of sucrose by fructosyltransferases. They help to promote the absorption of minerals in the intestine, inhibit the growth of harmful bacteria in the gut, and reduce cholesterol levels [29]. A population intervention study on nonalcoholic steatohepatitis showed that the food composed of fructan and *Bifidobacterium* can improve the composition of human fecal flora [30]. FOS have been considered to have bifidogenic effects and enhance the production of butyrate [31]. FOS are not broken down by small intestinal glycosidases and hence the structure is unaltered until they reaches the cecum. In the cecum, the gut microbiota metabolize FOS into hydrogen, SCFAs, carbon dioxide, L-lactate, and other metabolites [32]. FOS can increase the amount of *Lactobacilii* and *Bifidobacteria* [31]. FOS are transported across membranes by two phosphotransferase systems (SacPTS1 and SacPTS2), and then hydrolyzed in the cytoplasm by β-fructofuranosidase (SacA) [18] (Figure 1) (Table 1). It is worth mentioning that studies have shown that advanced glycation end products produced by fructose metabolism may be a burden to some patients with irritable bowel syndrome and may cause adverse consequences such as abdominal pain [33].

Inulin acts as a prebiotics in humans. The digestive enzymes in the human small intestine cannot hydrolyze inulin because it is a linear fructose polymer linked by β-2,1 glycosidic bonds instead of α-1,4 glycosidic bonds. However, *Bifidobacterium* and *Lactobacillus* can exploit it. Inulin has the potential to improve human intestinal flora and alleviate obesity [34,35]. Besides, inulin helps relieve constipation by lowering the pH level of the intestine and increasing the load or rate of stool, known as the bulking effect [36]. The concentration of gastrointestinal lactic acid bacteria will be changed due to inulin intake, thus stimulating T cell function and phagocytic activity. Inulin is hence believed to be beneficial to the immune system [37]. In addition, in the mouse tumor model, inulin diet intervention was found to increase the relative abundance of many intestinal bacteria with anti-tumor potential, such as *Bifidobacterium*, *Bacteroides*, *Parabacteroides*, *Akkermansia muciniphila*, and *Barnesiella*, further showing restrictions on the growth of colon cancer and melanoma tumor [38]. However, the effect of inulin on an organismic level remains to be discussed. Researchers found that feeding wild type mice with a high-fat diet rich in inulin leads to the development of dysplasia and hepatocellular carcinoma [39], which indicated that inulin may interact with other nutrients and deserves further study.

### 2.3. Starch and Cellulose

Starch is used as a storage of energy by most of the green plants and serves as the most familiar form of carbohydrate in the human diet. However, resistant starch is a type of prebiotics, which cannot be hydrolyzed by endogenous enzymes in the small intestine and it can lead to fermentation reactions for its utilization by intestinal microbiota in human gastrointestinal tract, especially in the colon [40]. It is estimated that about 20 g of resistant starch interacts with bacteria every day in the human lower intestine [41]. Amylolytic gut microbiota (e.g., *Firmicutes*, *Bacteroidetes*, and *Actinobacterium*), butyrogenic bacteria and methanogenic archaea ferment resistant starch, thus producing gas (e.g., methane, hydrogen, carbon dioxide), SCFAs (e.g., acetate, propionate, butyrate and valerate), organic acids (e.g., lactic, succinic and formate), branched SCFAs (e.g., isobutyrate and isovalerate) and alcohols (methanol and ethanol). These fermentation processes are completed by degrading starch polymers into glucose through glycolysis and methanogenic bacteria produce methane from formate, hydrogen and carbon dioxide (produced by bacteria that metabolize resistant starch) [41]. Adequate intake of resistant starch (≅20 g/day) [40] has the potential to prevent and treat some human diseases, including obesity [42], chronic kidney disease [2], cancer and hypertension [28,43,44].

Hundreds to thousands of D-glucoses are linearly linked by β-1,4 glycosidic bonds, forming cellulose. Cellulose is in the cell wall of greeneries and a variety of algae and it is also an extracellular polysaccharide widely found in biofilms [45]. In humans, cellulose is an important dietary fiber, which is insoluble. Since there is no cellulase in human digestive tracts, cellulose cannot be digested though it has an impact on health. On the one hand, it causes changes in the composition of gut microbiota [46]. In rats, high cellulose intake markedly increased the amount of *Actinobacteria* while the abundance of *Tenericutes* decreased when compared to low cellulose intake [46]. On the other hand, cellulose can be beneficial to intestinal health. Decreased consumption of cellulose is a crucial reason for the emergence of inflammatory bowel diseases, including Crohn’s disease and ulcerative colitis. A study has confirmed that the intake of cellulose during the pediatric period plays a great role in improving colitis in mice [46].

## 3. Dietary Amino Acids

### 3.1. Tryptophan and its Metabolites

Daily foods like eggs, liver, chicken and pork are good sources of AAA, such as tryptophan, tyrosine, and phenylalanine (Figure 1). Tryptophan metabolism is a systemic process, involving the intestinal tract, immune cells, brain and liver [47]. About 95% of the ingested tryptophan will enter the kynurenine pathway [47] and will be metabolized into a series of metabolites like kynurenic acid, picolinic acid and quinolinic acid, and finally produce nicotinamide adenosine dinucleotide [48]. A recent study found that the expression level of indoleamine 2,3-dioxygenase 1 (IDO1) was increased significantly in tumor tissues [49]. In addition, the increasing level of intestinal IDO can be correlated with obesity since it leads to altered tryptophan catabolism and the production of interleukin-22 [50]. When the activity of IDO was inhibited, the sensitivity of insulin improved, and chronic inflammation decreased [50]. The other bioactive compounds from tryptophan made by gut microbiota contain indole (metabolized by tryptophanase-expressing bacteria), indole-3-carboxaldehyde (I3A) (metabolized by *Lactobacilli*), IPA, tryptamine, indole sulfate, skatole, and so on (Figure 1). For a long time, it has been believed that only two bacteria from gut microbiota, namely *Clostridium sporogenes* and *Clostridium botulinum*, have the ability to metabolize indole into IPA, which appears to be metabolized by tryptophan deaminase [51,52]. However, a recent study revealed another four IPA-producing bacteria from gut microbiota, such as *Peptostreptococcus anaerobius* CC14N, *Clostridium cadaveries* CC88A, *Clostridium cadaveries* CC44 001G and *Clostridium cadaveries* CC40 001C [21]. Some microbiota and genes/enzymes related to the catabolism of tryptophan have been summarized in Table 1.

Tryptophan and its metabolites have been shown to exert two-sided effects on human health. The reduction in tryptophan level is associated with the progression of cachexia and weight loss, whereas tryptophan intake was negatively correlated with depression and positively correlated with sleep duration [53,54]. At the same time, indole helps to ensure the ability of commensal bacteria to resist pathogens in the gut, as a ligand for the aryl hydrocarbon receptor (AhR) and as a glucagon-like peptide-1 secretagogue [55]. In the presence of indole, IPA binds to pregnane X receptors specifically in the intestinal cells, thus maintaining the homeostasis of mucosal and barrier function [56]. Followed by the absorption from the intestine and distribution to the brain, IPA acts as a neuroprotective agent against cerebral ischemia and Alzheimer’s disease [57]. Tryptophan is metabolized by *Lactobacillus* species, and I3A is produced in the intestine, which also acts on AhR in the immune cells, thus increasing the production of IL-22 [58].

While indole, I3A and IPA appear to be beneficial, indole sulfate and skatole are found to be toxic [59]. Higher concentrations of indole sulfate are not only poisonous but also associated with vascular and renal diseases [60]. It is found that the oral charcoal adsorbent AST-120 binds indole in the intestinal lumen, and reduces the blood plasma indole sulfate levels, in turn decreasing kidney damage and atherosclerosis-related kidney injuries [61]. Another important product of tryptophan metabolized by mammalian gut microbiota is skatole, which has been proven to cause pulmonary edema in sheep, goats, and some strains of mice [62]. It selectively targets club cells, which are the key sites of cytochrome P450 enzymes in the lungs [62]. With the help of these enzymes, skatole is metabolized into 3-methyleneindolenine, which causes cell injury by forming protein adducts [62].

### 3.2. Tyrosine, Phenylalanine

In the liver, tyrosine is the metabolite of phenylalanine from the first irreversible step, and then it catabolizes into hydroxyphenylpyruvic acid, homogentisnic acid, maleylacetoacetate, fumarylacetoacetate, and so on [63]; however, the catabolism of tyrosine and phenylalanine within gut microbiota is different (Figure 1).

Phenyllactic acid (PLA) is produced during phenylalanine metabolized by *Lactobacillus* spp. [64]. PLA possesses a broad spectrum of antioxidant, antibacterial and antifungal activity [65]. Using scanning electron microscopy and fluorescence assays, researchers identified damages caused by PLA as morphological changes in *Listeria monocytogenes* and *Escherichia coli*. It seems that PLA could interact with bacterial genomic DNA by intercalation, which suggests that PLA possessed dual antibacterial targets including membrane and genomic DNA [65]. Besides, β-phenylethylamine is biosynthesized from L-phenylalanine by amino acid decarboxylases and many species including *Pseudomonas*, *Clostridium*, *Lactobacillus*, and *Enterobacteriaceae* [66], and it acts as a potent anti-microbial agent against certain pathogenic strains of *Escherichia coli* and as a treatment for depression [66].

*p*-cresol is one of the metabolites from tyrosine. However, it is reported that *p*-cresol has cytotoxic effects in renal tubular cells, thus causing a reduction in cellular activity and cell apoptosis [67]. Researchers have observed that even small doses of *p*-cresol can inhibit the proliferation and respiration of colonic epithelial cells, and higher doses increase the risk of DNA damage [68]. It is demonstrated that *p*-cresol prevents endothelial progenitor cells from proliferating through the p38 pathway [69]. *p*-cresol also disturbs cardiomyocyte adherens junction through promoting the synthesis of intercellular gaps [70].

### 3.3. Other Amino Acids

The following amino acids such as glutamine/glutamate, asparagine/aspartate, lysine, serine, threonine can also be consumed through daily foods and then catabolized by gut microbiota [71]. There are also some metabolites from aliphatic amino acids that are equally prominent but poorly characterized, such as δ-amino valeric acid [72], α-amino butyric acid [57], γ-amino butyric acid (GABA) [72] and 3-aminoisobutyric acid [73]. δ-amino valeric acid is biosynthesized from arginine, proline, and ornithine. α-amino butyric acid is derived from threonine or methionine, but its function has not yet been elucidated [57]. In addition, GABA is produced from glutamate, ornithine and arginine.

Some foods are a good source of glutamine, such as beef, fish, chicken, eggs, beans and especially milk. Apart from dietary intake, glutamine can also be synthesized from valine, leucine, isoleucine [74]. It helps to maintain the intestinal barrier, stimulates intestinal cells’ proliferation, and strengthens tight junctions [75]. A study showed that it helped to relieve irritable bowel syndrome, and to lower the scores assessed by the Irritable Bowel Syndrome Severity Scoring System [76]. When compared with L-alanine, L-glutamine decreases the ratio of *Firmicutes* to *Bacteroidetes* significantly, while this ratio increases in the L-alanine group. The *Firmicutes* to *Bacteroidetes* ratio is considered to be a biomarker for obesity [77].

GABA is a vital metabolite, which is produced by different kinds of gut bacteria like *Lactobacilli*, *Bifidobacterium*, *Bacteroides* and *Parabacteroides* [78,79]. It is reported that after Germ-free mice were inoculated with *Lactobacilli*, their memory improved, and higher levels of GABA in the hippocampus were observed [80]. As a neurotransmitter, it targets the brain directly by stimulating the vagus nerves or indirectly by the neuroendocrine mechanism, thus affecting appetite [81]. Besides, neurotransmitters, such as GABA and serotonin, also play an important role in the communication between microbiota and the immune system, and may affect the development of depression [82,83]. GABA is also evidenced to have connections with nervous system diseases like essential tremor, Huntington’s disease, and Parkinsonism [84]. Drugs or probiotics that target gut microbiota are under development and could be effective against those diseases, but their efficacy is still uncertain.

## 4. Dietary Lipids and Their Metabolites

Lipids are taken from food, including triacylglycerol, cholesterol, and phospholipids. At the intestinal tract, they are absorbed by intestinal mucosa under the action of bile acid and lipase, re-esterified by intestinal epithelial cells and packaged into chylomicrons in the endoplasmic reticulum, transported to the Golgi apparatus, and then secreted for transportation by the lymphatic system. Previous studies have shown that intestinal microorganisms can affect the lipid metabolism of the host. For example, some facultative and anaerobic bacteria in the large intestine produce secondary bile acid from bile salt pool secreted into the intestine. A small portion of bile acids derived from these bacteria are absorbed into the bloodstream and can regulate liver and/or systemic lipids and glucose metabolism through nuclear receptors or G protein-coupled receptors [85].

### 4.1. Fatty Acids

One of the metabolites of triacylglycerol is fatty acid. ω3 and ω6 essential polyunsaturated fatty acids profoundly affect the host immune system. In particular, lipid metabolites of ω3 essential polyunsaturated fatty acids (such as eicosapentaenoic acid and docosahexaenoic acid) have recently been proven to have anti-allergic and anti-inflammatory effects [86]. CLA are naturally occurring fatty acids found primarily in the milk and meat products of ruminants such as cattle and sheep. Moreover, they are a mixture of positional and geometric isomers of linoleic acid belonging to conjugated fatty acids that contain one or more cis-trans non-methylene discontinuous double bonds (Figure 1). Moreover, CLA can also be produced endogenously by the enzyme delta-9-desaturase [87]. The production of CLA isomers is mainly due to the rumen microbiota including the strains of *Clostridium proteoclasticum*, *Propionibacterium acnes*, *Butyrivibrio fibrisolvens*, *Megasphaera elsdenii*, and rumen protozoa [88]. The impact of CLA on host health is worth considering. They have a high activation potential of peroxisome proliferator-activated receptor gamma (PPAR-γ). Some studies have shown that dietary supplementation of CLA can activate PPAR-γ in macrophages and T cells and play anti-inflammatory and cancer-promoting roles. On the one hand, CLA can improve amyloid-induced colitis; in contrast, it can also promote the formation of colorectal cancer in mice [87]. When fed with 10-hydroxy-*cis*-12-octadecenoic acid, a metabolite derived from linoleic acid, NC/Nga mice had better health conditions, such as restoring skin barrier, suppressing skin inflammation, enhancing intestinal IgA production and regulating gut microbiota [89]. Although studies have shown that CLA have many effective physiological functions, such as preventing cancer, reducing weight, resisting diabetes and resisting hypertension [90], in-depth research is still needed before CLA could be widely recommended. Because CLA are a mixture of isomers, the efficacy of each isomer is different: the 9 cis(c), 11 trans(t) isomer is mainly to prevent cancer; the 10 t, 12 c isomer is to resist cancer, diabetes and lose weight [91].

### 4.2. Bile Acids and Cholesterol

Meat, animal liver and egg yolk are rich in cholesterol, and it is one of the oxidizable lipids composing membranes and plasma lipoproteins [92]. Moreover, cholesterol belongs to steroidal, which can constitute bile acids (Figure 1). It is exuded into the intestine through a complex multi-enzyme pathway to promote digestion, transportation, and absorption of nutrients [93]. Under physiological conditions, the gut rapidly reabsorbs bile acids and transports them to the liver via the portal circulation called enterohepatic circulation [94]. The bile salts from the gut microbiota in the small intestine, tauro-α-muricholic acid (T-α-MCA) and tauro-β-muricholic acid (T-β-MCA), are potent farnesoid X receptor (FXR)-antagonists, which suggest that the gut microbiota modulates bile acid synthesis by changing the bile acid pool composition and by alleviating FXR inhibition in the small intestine (Table 1). Furthermore, a new report shows that the gut microbiota can also influence colonic FOXP3+ regulatory T cells, which can express a transcription factor called RORγ, and the immunological homeostasis is related to gut microbiota through a pan-genomic biliary network interaction [95].

Initially, bile acids are thought to be helpful in the absorption of dietary lipids, but now they are widely accepted to be a vital regulator of metabolism and gut microbiota [96]. With respect to intestinal diseases, such as irritable bowel syndrome, short bowel syndrome, inflammatory bowel disease and *Clostridium difficile* infection, the changes of the composition of the gut microbiota and the content of bile acids are related. Similarly, some diseases outside of the intestine, for example, asthma and obesity, are also linked to them [97]. Since bile acids and cholesterol metabolism are inter-linked, a more effective recirculation of bile acids may lead to lower cholesterol metabolism rates. Gut microbiota metabolites not only have effects on health, but also seem to be a key regulator of the intestinal microbiota. In fact, the drop of bile acid levels in the intestine is related to gut microbiota overgrowth and inflammation [98]. As cirrhosis with bacterial dysbiosis is associated with low level of bile acids entering the gut, the feature of this dysbiosis is the obvious decrease in Gram-positive bacteria, such as *Blautia* and *Rumminococcaceae*, compared to the normal intestinal microbiota [99]. Another study demonstrates that increased levels of bile acids leads to significant inhibition of the *Bacteroidetes* and *Actinobacteria*, two of major phyla in the human bacteria community [100]. There are several bacteria that have the ability to degrade cholesterol, the substrate for bile acid synthesis, depending on the specific genes, for example, *ltp2*, encoding a lipid transfer protein, and *chsH1* and *chsH2*, being homologous with thiolases [24]. In *Mycobacterium tuberculosis*, the sequence of genes that constitute the degradation gene cluster of bile acids is the intracellular growth (*igr*) operon (Table 1). Low levels of cholesterol might increase the risk of premature birth, cancer, depression and anxiety; excessive cholesterol may increase the risk of stroke and coronary artery disease [100].

### 4.3. Choline

Choline is a positively charged four-valence base and the essential component of all biofilms and the precursor of acetylcholine in cholinergic neurons. Bovine milk contains a number of metabolic forms of choline, which contributes to the growth and development of the newborn [101]. Food rich in fats and cholesterol, like animal liver, egg yolk and red meat, often have important quantities of choline and L-carnitine, which are necessary for trimethylamine-*N*-oxide (TMAO) formation in the intestinal microbiota [102]. Many biological processes require choline, including protecting the structural integrity of the cell membranes, maintaining cholinergic nerve transmission, and having some synthetic reactions with methyl groups [103].

Furthermore, gut microbial metabolism of choline can produce trimethylamine (TMA) [102]. TMA is a colorless, toxic, and flammable tertiary amine compound and it is further oxidized to TMAO by flavin monooxygenase 3 or other flavanase secreted by the liver (Figure 1). TMAO enters the blood circulation, which is associated with some cardiovascular diseases such as atherosclerosis. According to a study, mice once fed with foods containing choline or directly fed with choline shortly after birth presented atherosclerotic plaques in the aortic root after 20 weeks. In addition, researchers found that the area of atherosclerotic plaque was positively correlated with the level of serum TMAO [104]. Furthermore, TMAO makes glucose tolerance worse, inhibits insulin signaling in the liver and promotes inflammation of adipose tissue [105]. Direct dietary exposure to TMAO and its precursors, like L-carnitine or choline, contributes to important reductions in reverse cholesterol transport *in vivo* and changes in cholesterol and sterol metabolic pathways in the liver, the artery wall and the intestines [102,106,107].

## 5. Different Dietary Patterns

### 5.1. Mediterranean Diet

The Mediterranean diet is a diet pattern that emphasizes plant food like fruits, vegetables, tree nuts and grains, olives as well as moderate fish intake [108]. Daily, it provides 43% of its energy from carbohydrates, 15% of its energy from protein and about 37% of its energy from total fat [109]. On the one hand, it has been proven to reduce the risk of cardiovascular diseases, colon cancer, lymphoma and fatty liver through intestinal microbial metabolites, like SCFAs [110,111,112,113,114]. Otherwise, it can increase intestinal microbial diversity and help alleviate depression symptoms [115]. Meanwhile, it can reverse the adverse effects on cardiovascular health caused by long-term exposure to air pollution, maybe resulting from the high content of antioxidants. The higher the Mediterranean diet index, the lower the risk of pollution-related deaths [116]. It has been shown that the Mediterranean diet can directly improve the composition of breast microbes, facilitating the synthesis of metabolites with anticancer effects, thus inhibiting tumorigenesis [117]. The Mediterranean diet also improves metabolic health and regulates intestinal flora in overweight and obese people. In an 8-week randomized controlled trial of 82 overweight and obese subjects, the Mediterranean diet intervention group customized their Mediterranean diet, based on daily energy intake. The results showed that after taking the Mediterranean diet, which is consistent with daily energy intake, the decrease in blood cholesterol and carnitine levels in plasma and urine was proportional to the compliance of the Mediterranean diet, and it led to a variety of changes in the microbiome and metabolic group, such as increasing dietary fiber-degrading *Bacillus przewalskii* [118]. Moreover, the Mediterranean diet may help increase the success ratio of *in vitro* fertilization treatment [119].

Polyphenols are worth mentioning. They are a category of secondary metabolites naturally found in plants, which are composed of more than one phenol unit. It is well known that red wine, dark chocolate, tea, and berries are some of the best-known sources. There are more than 8000 types of polyphenols have been isolated and identified. According to structure of polyphenols, they can be roughly divided into flavonoids, stilbenes, phenolic acids and lignans [120]. The bioavailability of many polyphenols is low, the result being that most of them would reach the colon completely. Dietary fiber components can keep their integrity in the upper digestive tract and be digested by bacteria in the colon. Besides that, dietary fiber is able to bind with polyphenols strongly and work synergistically with probiotics. Therefore, dietary fiber may assist polyphenols to reach the colon intact [121]. The colonic metabolites of polyphenols are usually well absorbed. Aromatic or phenolic acids with zero to three aromatic hydroxyl groups, and their mono- or di-methoxy analogues with a sidechain of one to five carbons, are the major catabolites produced from polyphenols by the microbiome [122]. Many of the health benefits associated with polyphenols are related to their role as antioxidants, which help reduce chronic inflammation, and have the potential to improve heart health [123]. Besides that, regular intake of polyphenols is thought to reduce blood sugar levels, contributing to a lower risk of type 2 diabetes. This is partly because polyphenols prevent starch from breaking down into monosaccharides, which reduces the rise in postprandial glucose levels [124]. One study showed that people who ate the most polyphenol-rich foods had a 57% lower hazard of developing type 2 diabetes over two to four years, compared with those who ate the lowest amounts [125]. Moreover, polyphenols may benefit digestion by promoting the growth of beneficial gut bacteria while fending off the harmful ones. For instance, a research suggests that polyphenol-rich tea extracts are able to promote the growth of *Bifidobacteria* [126].

In summary, the Mediterranean diet is generally considered to be the standard of healthy eating. But when giving dietary advice to the worldwide population, we need to take their cultural background and personal circumstances into account so that they are more likely to comply [127].

### 5.2. African Diet

In most of areas of Africa, meals usually consist of a mixture of vegetables, grains, legumes and fruits, which is consistent with local customs and climate. The African diet varies from place to place because of the differences in economic development and regional disparity (Figure 2). For example, the diet of a village in West Africa is high in starch and fiber but low in animal fat and protein, which means the diet is more similar to that of humans living in the time of the beginning of agriculture [128]. Specifically, the diet mainly includes cereals, vegetables and peas, and a small amount of meat will also be taken in the rainy season [128]. Researchers have found that different diets brought different compositions of gut microbes [129]. Diet has an immediate effect on the composition of the gut flora and creates a personalized gut microbiota [130]. The characteristics of rural Africans’ traditional diet are high in fiber and low in fat [131]. The intestinal microorganisms of native Africans are mainly *Prevotella*. Compared with African-Americans, there are more bacteria in the fecal samples of native Africans, and there are far more species of butyrate-producing microorganisms, more genes that encode methane and hydrogen sulfide production, and higher concentrations of SCFAs [132]. Metabolites such as butyrate help to reduce the risk of colon cancer, so rural Africa is a low risk area for colon cancer [132]. At the same time, gut microbiota also acts on food, affecting the digestion and decomposition of food, and has an impact on health as well [133].

### 5.3. Asian Diet

North Asia is mainly composed of Siberia and the northeastern edges of the continent, a part of Russia. Siberia is one of the coldest areas in the world and people living in North Asia need to adapt to the extremely cold climate, inconstant temperature and limited food [134]. Their diet greatly relies on cold weather crops and foods with high carbohydrates and saturated fats, which means it contains bread, fish, meat and some smaller amounts of fresh produce. A study found that some genes associated with lipid metabolism had changed in the Siberians because of the cold climate and eating habits; the variants connect to brown adipose tissue and they are helpful to understand some Siberians’ physiological differences [135]. In addition, some gut microorganisms, like *Prevotella* and *Bacteroides*, are found to lack dominance in North Asian people [136].

Next, we focus on East Asia. As one of the healthy diets, the traditional Japanese diet is based on rice, vegetables, fish and soybeans (Figure 2) [137]. The Okinawan diet, however, includes lots of sweet potatoes as a staple, instead of rice [138]. Theoretically, the Japanese diet would provide 452 kcal/100 g, 66.8% of which is from carbohydrate, another 12.6% from protein, and the other 20.6% from fat [139]. The staple food of the Chinese diet is also based on rice, wheat and highland barley are also included. In terms of meat intake, pork is a traditionally important meat source in East Asia. According to a Chinese diet study, in 2011, carbohydrate, protein and fat provide 54.3%, 13.3% and 32% of the energy, respectively [140]. In China, pork is the most consumed meat, followed by poultry. But in the traditional Japanese diet, compared with meat, fish and shellfish are more frequently consumed. Besides, soy products play an important role in both diets, which is a good source of plant protein. The Mongolian diet is rarely mentioned. According to a detailed study, the consumption of red meat and refined grains among Mongolian adults is high, while the consumption of fruits, nuts, fish is low. However, the consumption of milk and dairy products varies upon seasons and regions. It would provide 2069 to 3266 kcal/day, and carbohydrates account for 37.4%–43.8% of the energy supply per day, while protein and total fat provide 19.1%–22.4% and 36.0%–41.0% of energy, respectively [141].

The common feature of the South Asian diet is that it is rich in beans, which are important sources of fibers and proteins. In addition, eating curry is also the eating habit of people in South Asia. Curcumin, as the main ingredient of curry, has many benefits for humans. For example, it was proved that curcumin can increase the serum activity of antioxidants such as superoxide dismutase, thereby acting as an antioxidant [142]. However, compared with other populations, South Asians may have a higher risk of cardiovascular disease, which is closely related to their eating habits. South Asians, especially immigrants from South Asia to the United States, are increasingly overeating fried snacks, sweets and high-fat dairy products, which can do harm to their metabolism [143].

Central Asia is located in the heartland of Eurasia, and usually includes Kazakhstan, Kyrgyzstan, Tajikistan, Uzbekistan and Turkmenistan. For thousands of years, central Asia has been dominated by nomadic peoples. The Central Asians mainly believe in Islam, which forbids the consumption of pork, dead animals, and wild animals. On the basis of these dietary taboos, the central Asian countries have inherited the dietary habits of the nomadic peoples, with pasta, beef and mutton, milk, potatoes, onions, tomatoes and other vegetables as the main products, especially meat and whole milk. This dietary pattern includes high availability of animal-source foods and few plant foods, with risks of overweight and micronutrient deficiencies [141]. The microbiomes of central Asians and Europeans are quite different, with the most striking difference being significantly more samples falling within the *Prevotella*-rich enterotype, potentially reflecting regional diets and lifestyles [144].

### 5.4. Western Diet

The Western diet, marked by consumption of refined sugars, saturated fat and salt, may result in damaged heart, kidneys, and immune system [145]. The typical American diet would provide about 47% of its energy from carbohydrates, 16% of its energy from protein, and 36% of its energy from total fat [146]. However, the Western diet used in research may be different [147]. Since it is sometimes related to a high-fat diet, obesity seems to be an inevitable consequence [148]. For the immune system, dietary fats have the potential to change the lipids of the cell membranes and destroy the immune functions [149,150]. In the gut, they may cause a disruption of barriers and result in the leak of harmful substances into the blood stream, and immune dysfunction, which deteriorates infection [145,151,152]. Furthermore, it is suggested that the Western diet could influence a metabolic syndrome because it is related to a decrease in cognate gonadal steroids and the overdose of gonadal steroids in the opposite sex [148]. In recent years, it has been reported that high-fat diets led to the increase in bile acids and the amount of intestinal microbiota (*Clostridium*, *Bacteroides*, *Bifidobacterium* and *Lactobacillus*), which can influence bile salt hydrolase gene expression, leading to adverse consequences on colonic and host cardio metabolic health [153]. Moreover, high-fat diets can alter the *Lgr5*^+^ intestinal stem-cells in terms of quantity and function in mammalian intestine and improve the self-renewing ability of intestinal organoid by inducing a robust peroxisome proliferator-activated receptor delta [154]. It was found that feeding 400 μL of a 25% corn syrup (composed of fructose and glucose) solution to mice daily promoted intestinal tumor growth in mice with colon cancer, indicating that glucose and fructose have synergistic effects on promoting intestinal tumor formation [153]. Furthermore, glycotoxicity, which is associated with advanced glycation end products, excess fructose, artificial sweeteners and high temperature cooking, leads to rapid aging and various metabolic diseases [155]. Overall, the Western diet appears to be unhealthy, because the food is calorie-dense, highly nonhomogeneous of fat, with high carbohydrate and protein contents (Figure 2) [148]. More fruits and vegetables may be taken with the Western diet, like the traditional Japanese diet and the Mediterranean diet [156].

### 5.5. Paleo Diet

The Paleo diet is one of new dietary trends and it avoids grains, dairy and any other processed food [158,159]. Carbohydrates provide about 35.5% of the energy, while protein provides 21%, and about 45.5% of energy comes from total fat [160]. A study shows that the gut microbiota of a group from Tanzania called Hadza, whose diet pattern is similar to the Paleo diet, is more rich and diverse compared to the Mediterranean diet control group [158]. Hadza feces seem to have more hexoses, sphingolipids and glycerophosholipids and less amino acids and biogenic amines [158]. However, there is no conclusive and convincing evidence to show the Paleo diet’s positive effect on type 2 diabetes and other metabolic diseases [159]. Furthermore, since this diet does not include whole grains, the intake of resistant starch is reduced.

### 5.6. DASH

The full name of DASH is Dietary Approaches to Stop Hypertension. As the name implies, it could help treat hypertension. The core of the DASH diet is to reduce the intake of sodium, sugar and red meat while increasing the intake of potassium, magnesium, calcium, protein, and fiber. It mainly includes fruits, vegetables, low-fat milk, whole grains, fish, poultry, beans, nuts, and other foods. About 55% of the energy is provided by carbohydrates, 17% by protein and another 27% by total fat [161]. This diet is good for the heart because it limits saturated fat and trans-fat and is rich in nutrients that help control blood pressure. The more consistent finding is that it can reduce the blood pressure of hypertension patients, meaning that DASH can improve other diseases with hypertension complications, such as gout accompanied by hypertension [162]. Other studies have pointed out that the DASH diet prevents the development of diabetes and nephropathy. A meta-analysis shows that adhering to the DASH diet reduces the risk of colorectal cancer. Because of the numerous benefits of the DASH diet, some people regard it as the healthiest diet along with the Mediterranean diet. The potential adverse effects, such as flatulence, may be caused by vegetable foods such as whole grains and vegetables, which contain a large amount of fiber. Since the DASH diet has a very detailed meal plan, it may be more complicated to implement.

### 5.7. Optavia Diet

The Optavia diet is a low-calorie, low-carb diet that combines packaged foods with homemade diets to achieve the goal of weight loss. It includes homemade meals known as “Lean and Green” and “Optimal Health Fuelings”, including food forms like sticks (cereal bars, etc.), cookies, milkshakes, pudding, cereals (cereal, cornflakes, etc.), soup and pasta (spaghetti, macaroni, etc.) [163]. The main features of the Optavia diet are low carbohydrates and high protein, and it is rich in probiotics, which can improve intestinal health.

In order to lose weight better, the Optavia diet has set up different diet plans. For example, the popular “Optimal weight 5&1 program”, consists of five “Optimal Health Fuelings” meals and a balanced “Lean and Green” diet. A 16-week study of 198 overweight or obese people found that the weight, fat level and waistline of the Optavia “Optimal weight 5&1 program” group were significantly lower than those of the control group [164]. More research is, however, needed to assess the long-term effects.

### 5.8. Ketogenic Diet

The Ketogenic diet is a high-fat, low-carb diet, and a suitable formula for protein and other nutrients. The classic Ketogenic diet mainly contains long-chain fats, which provide 60%–80% dietary energy [165]. In mouse models, it was proven that the antiepileptic effect of the ketogenic diet was mediated by intestinal flora, and the ketogenic diet played a key role in enriching *Akkermansia* and *Parabacteroides*. The interaction of these two bacteria can reduce the activity of gamma-glutamyl transpeptidase in intestinal flora, and thus have anti-seizure effects [166]. In the treatment of cancer, glucose–insulin feedback caused by PI3K inhibitors activates the PI3K-MTOR signaling pathway in the tumor. Moreover, the Ketogenic diet can improve the efficacy of PI3K inhibitors by inhibiting the disorder of glucose–insulin feedback [167]. The medium chain triglyceride (MCT) diet is also a ketogenic diet model. The MCT diet uses medium chain fatty acids for a much smaller proportion of energy (45%), allowing for higher carbohydrate intake, and may be used to prevent Alzheimer’s disease, cancer, and diabetes [165]. The Ketonic diet family also includes the Modifified Atkins diet and Low glycemic index treatment, for they use fat as the major energy supply [168]. However, the Ketogenic diet may be inadequate for dietary fiber and macronutrients, and its long-term effects, safety, and health effects remain to be investigated [169].

## 6. Conclusions

Human health is determined by various factors including the interactions between genes and environment. It is realized that the microbial genomes in the human body are constantly changing, and many factors would mediate these dynamic changes. They are the genes that are diversified in human beings themselves [170], and the internal and external environment of the human body, including the emotions [171], diet, air [172], and so on. Diet is a complex but relatively controllable factor when compared to emotions and air. Its complexity is reflected not only in the diversity of the chemicals in the diet substances themselves, but also in the diversity of the diet patterns and the diversity of the gut microbiota-derived metabolites from the diet sources. Gut microbiota also used ingredients in food for biosynthesis [173], which is not focused on here.

A better understanding of intestinal microbial metabolites of dietary sources will provide guidance to the daily diet of humans, which can promote human health and longevity. Many studies have confirmed that intestinal bacteria are associated with many diseases, like obesity, type 2 diabetes mellitus, angiocardiopathy, inflammatory bowel disease and chronic kidney disease [174]. Through diet intervention, intestinal microbial metabolites of dietary sources can be modulated, which can alleviate and treat many diseases to some extent. Taking prebiotic supplements scientifically, moderately, and duly has positive impacts on human health. For example, it is verified that inulin is capable of improving the function of vascular endothelium [175]. Nevertheless, it is also reported that ingesting inulin disrupted the balance of intestinal bacteria, resulting in cholestatic liver cancer in mice [39].

Nonetheless, with the accelerating pace of globalization, the diet structure is quietly undergoing worrying changes across the globe. For example, the Japanese diet has been westernized rapidly ever since Meiji Restoration in 1868 [176]. A study showed that between 2003 and 2015, the scores of the “plant food and fish” pattern decreased, while the scores of the “bread and dairy” pattern and the “animal food and oil” pattern increased, suggesting an increase in the ingestion of high-fat food and refined grain [177]. Meanwhile, the number of patients with irritable bowel syndrome in Japan has increased by 100 times compared with 30 years ago [176]. Connectedly, such changes of diet are also taking place in China. Before the 1980s, Chinese people rarely consumed a Western diet, and traditional Chinese food is composed of rice, tofu, fish, fermented sauces, and so on [178]. However, the percentage of energy consumed from fats increased from 10% to 32% in 9 provinces in 20 years [140]. Another study also found that more than 50% of Chinese people consumed excess oil and salt compared with the recommended dose allowance from Chinese Food Pagoda 2016 [179].

With the researches on intestinal microbial metabolites of dietary sources stepping forward, more and more metabolites will be discovered, which will eventually provide more guidance on the human diet and probiotic supplements. It is important to note that the mechanisms of diet–microbe–host interactions are quite complex and current models are mostly limited to animal models. Intestinal organoids may be a feasible model to overcome such issues [180]. Besides, the so-called optimal diet may not fit all individuals. Based on the forthcoming findings, we can develop personalized nutritional approaches as well as precision medicine [181,182], which may accelerate the recovery of many diseases and improve human health. Diet therapies that target gut microbiota have to be explored further [183].

## Figures and Tables

**Figure 1 foods-09-01045-f001:**
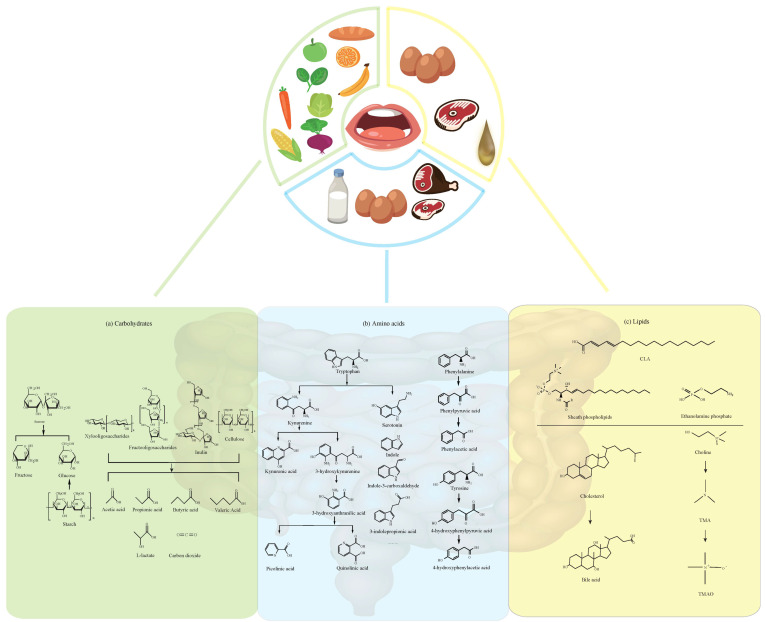
Foods that we consume are metabolized by the intestinal microbiota. Thousands of microbiotas in the gut participate in metabolism, breaking down the dietary substrates. (**a**) The metabolism of some important carbohydrates. Monosaccharide fructose can be directly absorbed in the intestinal tract. If it is ingested in the form of sucrose, it is hydrolyzed into glucose and fructose under the action of sucrase and then absorbed. Xylooligosaccharides are degraded into short chain fatty acids (SCFAs) as substrate of beneficial bacteria in the intestinal tract. Fructooligosaccharides (including inulin) are metabolized in the cecum by the intestinal microbiota to hydrogen, short-chain carboxylic acid, carbon dioxide, L-lactic acid, and other metabolites. Starch is hydrolyzed to glucose by amylase and then absorbed. Resistant starch is not digestible and is used by microorganisms in the colon. Cellulose cannot be digested. (**b**) In kynurenine pathway, tryptophan turns to kynurenine. Kynurenine will be catabolized to kynurenic acid by kynurenine amino-transferases, and to 3-hydroxykynurenine by kynurenine monooxygenase. 3-hydroxykynurenine will be catabolized to 3-hydroxyanthranilic acid, picolinic acid and quinolinic acid through a series of metabolic processes. In addition, tryptophan will be catabolized to serotonin, indole, and the other metabolites. At present, the known pathway that is mediated by gut microbiota is that phenylalanine converts into phenylpyruvic acid and phenylacetic acid, while tyrosine metabolites to 4-hydroxyphenylpyruvic acid and 4-hydroxyphenylacetic acid and finally *p*-cresol [8]. (**c**) Lipids are classified into cholesterol, linoleic acid and choline. Cholesterol is metabolized to bile acids by gut microbiota and linoleic acid is converted into conjugated linoleic acid (CLA). Choline becomes trimethylamine (TMA) firstly and trimethylamine-*N*-oxide (TMAO) in the further step.

**Figure 2 foods-09-01045-f002:**
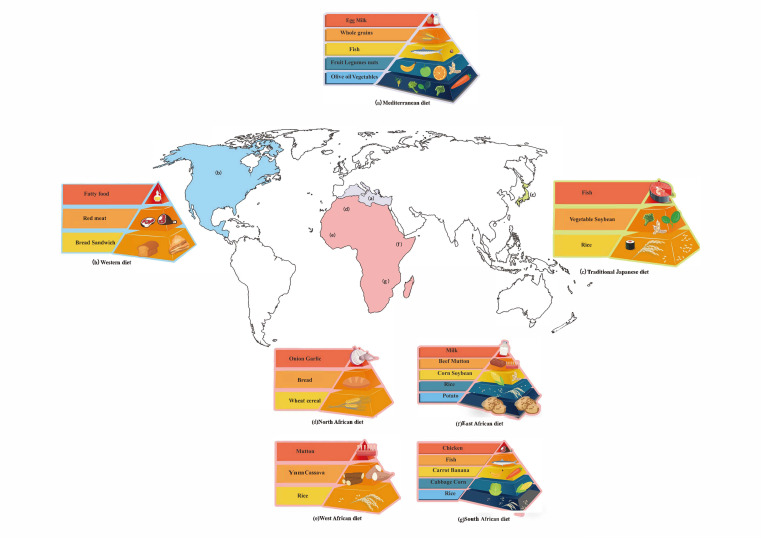
This figure illustrates the food habits in several representative areas of the world. (**a**) The most distinctive feature of the Mediterranean diet is that most people use olive oil as a source of fat. The diet is based on plant foods like carrots, bananas and oranges, with a small amount of fish, whole grains, milk, and eggs. (**b**) However, Westerners eat bread and sandwiches as staple food, and they prefer cheese. (**c**) Grains and rice are staples of the traditional Japanese diet. Vegetables and peas are often eaten in Japan. Because of developed fisheries, fish is common in the Japanese diet. (**d**) African diets are diverse and complex, with different regions having different eating habits and cultures. In North Africa, they eat grains and bread as staple foods, and onions and garlic are the most common vegetable. (**e**) Nonetheless, rice is a staple food of western Africans and cassava and sweet potatoes are the most unique elements of their diet [157]. (**f**) In East Africa, potatoes and rice are their staple food and corn and peas are also popular with the locals. Their main meat is beef and mutton and they also drink milk as a nutritional supplement. (**g**) Rice is the staple food in South Africa and cabbage, corn and carrots are quite popular there. South Africa is rich in bananas, so bananas are a common fruit. The main meats are fish and chicken because of geographical environment factors. In a nutshell, because of differences in religion and eating habits, each region has its own unique style of eating.

**Table 1 foods-09-01045-t001:** Typical dietary substrates metabolized by gut microbiota.

Substrate	Metabolite	Producer	Known/Predicted Activity	Gene/Enzyme	Reference
Xylooligosaccharides	Xylose Xylobiose	*Bacteroides*	As a unique carbon source	Polysaccharide Utilization Loci	[16]
Fructooligosaccharides	Fructose Glucose-6-P	*Lactobacillus plantarum* ST-III	Improves the composition of the intestinal microbiota by increasing the numbers of *Bifidobacteria* and *Lactobacilli*	Fructofuranosidase	[18]
Tryptophan	Indole	*Clostridia*	Helps to relieve Non-steroidal anti-inflammatory drug enteropathy	Not mentioned	[19]
		Not mentioned	Inhibiting *Acinetobacter baumannii*	*iifC*	[20]
	IPA	*Clostridium sporogenes*	Influencing host immunity	*fldC*	[21]
	Indole-3-lactic acid	*Lactobacillus reuteri*	Inducing gut intraepithelial CD4^+^CD8αα^+^ T cells	Aromatic aminotransferase	[22]
Bile acid	T-β-MCA	*Bacteroides, Clostridium, Lactobacillus, Bifidobacterium, Listeria*	Antagonist of farnesoid X receptor (FXR)	*ibabp*	[23]
Lipid	Cholesterol	*Mycobacterium tuberculosis,*	Cholesterol side chain degradation	*fadE28* and *fadE29* (within the *igr* operon); *chsH1* and *chsH2*; *ltp2*	[24]
		*Rhodococcus jostii,*			
		*Comamonas testosterone,*			
		*Pseudomonas sp.*			
	CLA	*Butyrivibrio fibrisolvens*, *Megasphaera elsdenii*, *Propionibacterium acnes* and *Clostridium proteoclasticum*, L. *plantarum* AKU 1009a	Anti-carcinogenic, anti-obesity, anti-cardiovascular disease and anti-diabetic activities	*cla-dh; cla-dc; cla-er*	[25]

IPA: 3-indolepropionic acid; T-β-MCA: tauro-β-muricholic acid; CLA: conjugated fatty acids.

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
