# Peer review of "Dietary Components, Microbial Metabolites and Human Health: Reading between the Lines"

_foods, 2020, doi:10.3390/foods9081045_

Round 1

Reviewer 1 Report

The title of this review is sound but I do not see the parto on gut microbiota, please add it or change the title.

The English style and grammar has to be extensively reviewed.

The figures are appropriate and well done.

Author Response

Point 1: The title of this review is sound but I do not see the parto on gut microbiota, please add it or change the title.

Response 1: Thanks for your careful review. The first part of this manuscript focus on how gut microbiota metabolizes nutrients. And gut microbiota-derived metabolites are also discussed. According to your suggestion, we have changed the title as “Dietary Components, Microbial Metabolites and Human Health: Reading between the Lines” to avoid ambiguity (in Red).

Point 2: The English style and grammar has to be extensively reviewed.

Response 2 Thanks for your advice. The manuscript has been checked extensively for English and grammar by a native speaker (in Red).

Reviewer 2 Report

Manuscript ID: foods-855469

Title: Dietary components, intestinal microbial and human health: Reading between the lines    

This paper is a review of the effects of dietary ingredients on intestinal bacteria, as well as dietary patterns and health in each region. It was difficult to understand how the description of the relationship between dietary components and intestinal bacteria in the first half relates to the relationship between dietary patterns and health in each region in the second half. It was thought that understanding would be deepened if the dietary patterns and the amounts of ingredients in each region were also added. And, In addition, in the relationships between dietary components and intestinal flora, it has been recently reported that polyphenols affect intestinal flora, so it seems necessary to add them.

Author Response

Point 1: This paper is a review of the effects of dietary ingredients on intestinal bacteria, as well as dietary patterns and health in each region. It was difficult to understand how the description of the relationship between dietary components and intestinal bacteria in the first half relates to the relationship between dietary patterns and health in each region in the second half. It was thought that understanding would be deepened if the dietary patterns and the amounts of ingredients in each region were also added.

Response 1: We are pleased that the reviewer made constructive suggestions. We have added the relevant content in line 246, 286 290, 297, 323, 336, 363, 371 (in Red).

Point 2: And, In addition, in the relationships between dietary components and intestinal flora, it has been recently reported that polyphenols affect intestinal flora, so it seems necessary to add them.

Response 2: Thanks for your advice. We have revised and added polyphenols in line 398-419 (in Red).

Reviewer 3 Report

In presented review authors have summarized the possible molecular mechanisms by which commonly consumed dietary components are metabolism by our gut microbiota. It is relevant and interesting, the detailed dietary component metabolism by gut microbiota, which is not present in other published review. The paper well written, the text is clear and easy to read and need minor English editing. The conclusions are consistent with the evidence and arguments presented.

Reviewer has minor edits to further improve the quality of the drafted review article.

  1. Add a keto-diet section and discuss the significance in review article context.
  2. Expand GABA and CLA in review.

Author Response

Point 1: In presented review authors have summarized the possible molecular mechanisms by which commonly consumed dietary components are metabolism by our gut microbiota. It is relevant and interesting, the detailed dietary component metabolism by gut microbiota, which is not present in other published review. The paper well written, the text is clear and easy to read and need minor English editing. The conclusions are consistent with the evidence and arguments presented.

Response 1: We are very pleased that the reviewer made encouragement at this manuscript. Moreover, the English of this manuscript has been polished by a native speaker (in Red). 

Point 2: Reviewer has minor edits to further improve the quality of the drafted review article.

Add a keto-diet section and discuss the significance in review article context.

Response 2: We have revised, and keto-diet has been added to the section in line 360-375 (in Red).

Point 3: Expand GABA and CLA in review.

Response 3: Thanks for your advice, we have expanded relevant content in line 145-153 and line 183-188 (in Red).

Round 2

Reviewer 1 Report

The English form has been improved and the work results more readable. 

However, there are still mistakes such as the use of singular for plural subjects, e.g. line 147. Please revise with attention throughout the paper.

Another issue is the lack of description of the homeland main diet, that of Asian continent. Could you please add it ? and discuss properly the differences with the others.

Finally, why the discussion over polyphenols is attached to the conclusions ? Please remove and adapt to the nutrients preliminary section, isn't it more appropriate ?

Beside these changes, the manuscript can be finally revised. 

Author Response

Point 1:However, there are still mistakes such as the use of singular for plural subjects, e.g. line 147. Please revise with attention throughout the paper.

Response 1:Thanks for your advice. We have revised carefully throughout the paper.

Point 2:Another issue is the lack of description of the homeland main diet, that of Asian continent. Could you please add it ? and discuss properly the differences with the others.

Response 2:We are pleased that the reviewer made constructive suggestions. We have added the relevant content in line 307-349(in Blue)

Point 3:Finally, why the discussion over polyphenols is attached to the conclusions ? Please remove and adapt to the nutrients preliminary section, isn't it more appropriate ?

Response 3:Thanks for your advice. We have moved this part to Mediterranean diet, line 264-285 (in Blue), because Mediterranean diet has high phenolic-rich foods intake.

Reviewer 2 Report

Manuscript ID: foods-855469   I believed that this paper was useful for the researcher of  foods and nutritions.   Sincerely yours,

Author Response

Thank you very much!